# Ensemble learning for Physics Informed Neural Networks: a Gradient Boosting approach

## Abstract

While the popularity of physics-informed neural networks (PINNs) is steadily rising, to this date, PINNs have not been successful in simulating multi-scale and singular perturbation problems. In this work, we present a new training paradigm referred to as "gradient boosting" (GB), which significantly enhances the performance of physics informed neural networks (PINNs). Rather than learning the solution of a given PDE using a single neural network directly, our algorithm employs a sequence of neural networks to achieve a superior outcome. This approach allows us to solve problems presenting great challenges for traditional PINNs. Our numerical experiments demonstrate the effectiveness of our algorithm through various benchmarks, including comparisons with finite element methods and PINNs. Furthermore, this work also unlocks the door to employing ensemble learning techniques in PINNs, providing opportunities for further improvement in solving PDEs.

## 1 Introduction

Physics informed neural networks have recently emerged as an alternative to traditional numerical solvers for simulations in fluids mechanics Raissi et al. (2020); Sun et al. (2020), bio-engineering Sahli Costabal et al. (2020); Kissas et al. (2020), meta-material design Fang & Zhan (2019); Liu & Wang (2019), and other areas in science and engineering Tartakovsky et al. (2020); Shin et al. (2020). However, PINNs using fully connected, or some variants architectures such as Fourier feature networks Tancik et al. (2020), fail to accomplish stable training convergence and yield accurate predictions at whiles, especially when the underlying PDE solutions exhibit high-frequencies or multi-scale features Fuks & Tchelepi (2020); Raissi (2018). To mitigate this pathology, Krishnapriyan et al. (2021) proposed a sequence-to-sequence learning method for time-dependent problems, which divide the time domain into sub-intervals and solve the problem progressively on each them. This method avoids the pollution of the underlying solution due to the temporal error accumulation. Wang et al. (2022) elaborated the reason that the PINNs fail to train from a neural tangent kernel perspective, and proposed an adaptive training strategy to improve the PINNs' performance. An empirical learning-rate annealing scheme has been proposed in Wang et al. (2021), which utilizes the back-propagated gradient statistics during training to adaptively assign importance weights to different terms in a PINNs loss function, with the goal of balancing the magnitudes of the gradients in backward propagation. Although all of these works were demonstrated to produce significant and consistent improvements in the stability and accuracy of PINNs, the fundamental reasons behind the practical difficulties of training fully-connected PINNs still remain unclear Fuks & Tchelepi (2020).

Besides PINNs, many other machine learning tasks suffer from the same issues, and some of these issues have been resolved by gradient boosting method. The idea of gradient boosting is blending several weak learners into a fortified one that gives better predictive performance than could be obtained from any of the constituent learners alone Opitz & Maclin (1999). For example, Zhang & Haghani (2015) proposes a gradient-boosting tree-based travel time prediction method, driven by the successful application of random forest in traffic parameter prediction, to uncover hidden patterns in travel time data to enhance the accuracy and interpretability of the model. Callens et al. (2020) used gradient boosting trees to improve wave forecast at a specific location whose RMSE values in average 8% to 11% lower for the correction of significant wave height and peak wave period. Recently, many researchers have contributed to gradient boosting method

and further improved its performance. Friedman (2002) shows that both the approximation accuracy and execution speed of gradient boosting can be substantially improved by incorporating randomization into the procedure, and this randomized approach also increases robustness against overcapacity of the base learner. Ke et al. (2017) found that the efficiency and scalability of Gradient Boosting Decision Tree (GBDT) are unsatisfactory when the feature dimension is high and data size is large and a greedy algorithm has been used to effectively reduce the number of features without hurting the accuracy of split point determination by much and thus solve the issue.

Inspired by the above-mentioned literature review, we arrive at our proposed method in this paper. In this work, we present a gradient boosting physics informed neural networks (GB PINNs), which adopts a gradient boosting idea to approximate the underlying solution by a sequence of neural networks and train the PINNs progressively. Specifically, our main contributions can be summarized into the following points:

1. Inspired by the GB technique prevalent in traditional machine learning, we introduce a GB PINNs approach. This addresses many of PINNs' shortcomings, including issues related to sharp gradients.

2. Our method's enhanced performance is justified from various perspectives. Initially, drawing parallels with traditional gradient boosting, we deduce that a composite learner outperforms isolated ones. Furthermore, leveraging the neural tangent kernel theory, we demonstrate that our approach effectively mitigates lazy training challenges.

3. To substantiate our techniques, we present several rigorous experiments, encompassing even those in numerical analysis. We provide an ablation study and detail the trade-offs between time and memory, aiding readers in comprehending the practical implications of our method.

4. In our research, we not only corroborate the efficacy of Fourier feature networks through extensive numerical experiments—as previously established in related work Wang et al. (2022; 2021); Tancik et al. (2020)—but also extend these findings in a meaningful way. Specifically, our ablation studies reveal that our GB PINNs framework, which integrates Fourier features as a key component, consistently outperforms standalone Fourier feature networks. This result underscores the unique synergistic advantages of combining GB PINNs with Fourier features, thereby making a novel contribution to the existing body of literature.

We introduce some preliminaries for key ingredients of our algorithm in section 2. Then we present our algorithm with motives in section 3. Numerical experiments are shown in section 4 to verify our algorithm. We discuss our algorithm and conclude the paper in section 5.

## 2 Preliminaries

In this section, we will provide a brief overview of the related topics that are relevant to the proposed algorithm in this paper. For a more in-depth understanding of these topics, we encourage readers to refer to the original papers cited below.

### 2.1 Physics informed neural networks

In this subsection, we provide a brief overview of PINNs. For a more comprehensive introduction to PINNs, readers are directed to Raissi et al. (2019). PINNs are a method for inferring a continuous latent function $u(\boldsymbol{x})$ that serves as the solution to a nonlinear PDE of the form:

$$\mathcal{N}[u](\boldsymbol{x}) = 0, \quad \text{in } \Omega, \tag{1}$$

$$\mathcal{B}[u](\boldsymbol{x}) = 0, \quad \text{on } \partial\Omega, \tag{2}$$

where $\Omega$ is an open, bounded set in $\mathbb{R}^d$ with a piecewise smooth boundary $\partial\Omega$, $\boldsymbol{x} \in \mathbb{R}^d$, and $\mathcal{N}$ and $\mathcal{B}$ are nonlinear differential and boundary condition operators, respectively.

The solution to the PDE is approximated by a deep neural network, $u_\theta$, which is parameterized by $\theta$. The loss function for the network is defined as:

$$L(u;\theta) = \frac{\omega_e}{N_p} \sum_{i=1}^{N_p} |\mathcal{N}[u_\theta](\boldsymbol{x}_i^p)|^2 + \frac{\omega_b}{N_b} \sum_{i=1}^{N_b} |\mathcal{B}[u_\theta](\boldsymbol{x}_i^b)|^2, \tag{3}$$

where $\{\boldsymbol{x}_i^p\}_{i=1}^{N_p}$ and $\{\boldsymbol{x}_i^b\}_{i=1}^{N_b}$ are the sets of points for the PDE residual and boundary residual, respectively, and $\omega_e$ and $\omega_b$ are the weights for the PDE residual loss and boundary loss, respectively. The neural network $u_\theta$ takes the coordinate $\boldsymbol{x}$ as input and outputs the corresponding solution value at that location. The partial derivatives of the $u_\theta$ with respect to the coordinates at $\mathcal{N}$ in equation 3 can be readily computed to machine precision using reverse mode differentiation Baydin et al. (2018).

The loss function $L(u;\theta)$ is typically minimized using a stochastic gradient descent algorithm, such as Adam, with a batch of interior and boundary points generated to feed the loss function. The goal of this process is to find a set of neural network parameters $\theta$ that minimize the loss function as much as possible.

It is worth noting that the abstract PDE problem in equation 1-equation 2 can easily be extended to time-dependent cases by considering one component of $\boldsymbol{x}$ as a temporal variable. In this case, one or more initial conditions should be included in the PDE system and additional initial condition constraints should be added to the loss function 3.

## 2.2   Gradient boosting machines

In this subsection, we offer a concise overview of GB for traditional machine learning tasks. For a thorough exploration of this topic, refer to Hastie et al. (2009). Gradient Boosting (GB) is a powerful machine learning technique that is commonly used in regression and classification tasks. It is an additive ensemble of weak prediction models, similar to AdaBoost, but with a key difference - unlike other ensemble algorithms, GB does not have trainable weights, and the sub-models are trained sequentially instead of in parallel. For the sake of simplicity, in the rest of the paper, we will use $f(x;\theta)$ to denote a general neural network $f$ with input $x$ and parameterized by $\theta$.

Given a neural network $f(x;\theta)$ and a training dataset, the loss function $L(f;\theta)$ is defined as the sum of the individual losses for each sample, as follows:

$$L(f;\theta) = \sum_{i=1}^{N} L(y_i, f(x_i;\theta)),$$

where $N$ is the total number of samples in the dataset, $y_i$ is the true label for sample $x_i$, and $f(x_i;\theta)$ is the predicted label for sample $x_i$.

To minimize this loss function, a common approach is to use the stochastic gradient descent algorithm. This algorithm updates the network's parameters, $\theta$, iteratively using the following update rule:

$$\theta \leftarrow \theta - \gamma \frac{\partial}{\partial \theta} L(f;\theta), \tag{4}$$

where $\gamma$ is a user-specified learning rate that controls the step size of the updates.

The goal of the gradient boosting (GB) method is to minimize the loss function $L(f;\theta)$ with respect to the neural network function $f$. GB method assumes that the surrogate model can be represented in the following iterative form:

$$f_m(\boldsymbol{x};\Theta_m) = f_{m-1}(\boldsymbol{x};\Theta_{m-1}) + \rho_m h_m(\boldsymbol{x};\theta_m), \quad \text{for } m = 1, 2, 3, \cdots, \tag{5}$$

where $f_0(\boldsymbol{x};\theta_0)$ is a pre-selected baseline neural network, $\rho_m$ is the learning rate, $f_m(\boldsymbol{x};\Theta_{m-1})$ is parameterized by $\Theta_m = \bigcup_{i=0}^{m} \theta_i$, and $h_m(\boldsymbol{x};\theta_m)$ is a neural network designed to enhance the accuracy of the predictor $f_{m-1}(\boldsymbol{x};\Theta_{m-1})$. The gradient descent algorithm is used to choose $h_m(\boldsymbol{x};\theta_m)$, which is defined as:

$$h_m(\boldsymbol{x};\theta_m) = -\frac{\partial}{\partial f_{m-1}(\boldsymbol{x};\Theta_{m-1})} L(f_{m-1};\Theta_{m-1}). \tag{6}$$

Therefore, the model update rule is defined as:

$$f_m(\boldsymbol{x}; \Theta_m) = f_{m-1}(\boldsymbol{x}; \Theta_{m-1}) - \rho_m \frac{\partial}{\partial f_{m-1}(\boldsymbol{x}; \Theta_{m-1})} L(f_{m-1}; \Theta_{m-1}). \tag{7}$$

In this fashion, the corresponding loss at the $m$-th step reads

$$L(f_m; \Theta_m) = L(f_{m-1} + \rho_m h_m; \Theta_m). \tag{8}$$

The technique outlined in this construction is commonly referred to as a GB method. The update function, $h_m(\boldsymbol{x}; \theta_m)$ in equation 6, is similar in nature to the gradient vector in equation 4, however, GB operates by taking the gradient with respect to the function, rather than the parameter vector as traditional gradient descent does. This distinction is the reason why we refer to GB as a method that descends the gradient in function space. For further information on gradient boosting methods, please refer to the reference Hastie et al. (2009). Furthermore, it is worth noting that in the context of PINNs, this method has been adapted to a simpler form that is easily implementable.

## 3 Gradient boosting physics informed neural networks

Despite a series of promising results in the literature Hennigh et al. (2021); Kissas et al. (2020); Raissi (2018); Raissi et al. (2020); Sun et al. (2020), the original formulation of PINNs proposed by Raissi et al. (2019) has been found to struggle in constructing an accurate approximation of the exact latent solution. While the underlying reasons remain largely elusive in general, certain failure modes have been explored and addressed, as evidenced in Wang et al. (2021; 2022); Krishnapriyan et al. (2021). However, some observations in the literature can be used to infer potential solutions to this issue. One such observation is that the prediction error in PINNs is often of high frequency, with small and intricate structures, as seen in figures 4(b) and 6(a) and (b) of Wang et al. (2022). As demonstrated in Tancik et al. (2020), high-frequency functions can be learned relatively easily using Fourier features. Based on these findings, it is natural to consider using a multi-layer perceptrons (MLPs) as a baseline structure in PINNs, followed by a Fourier feature network, to further minimize the error. This idea led to the development of GB PINNs.

### 3.1 Introduction of GB PINNs and mathematical formulation

The proposed method, referred to as GB PINNs, utilizes a sequence of neural networks in an iterative update procedure to gradually minimize the loss function. As shown in equation 6, the update model $h_m(\boldsymbol{x}; \theta_m)$ is defined by the gradient of the loss with respect to the previous output $f_{m-1}$. However, in the context of PINNs, the PDE residual loss in equation 3 typically includes gradients of the outputs with respect to the inputs. This necessitates the computation of twisted gradients, which is a unique characteristic of this approach. For example

$$\frac{\partial}{\partial f(\boldsymbol{x}; \theta)} \left[ \left( \frac{\partial f(\boldsymbol{x}; \theta)}{\partial \boldsymbol{x}} \right)^2 \right],$$

which is definitely not elementary and should be avoided. Despite the mathematical validity of the gradient

$$\frac{\partial}{\partial f_{m-1}(\boldsymbol{x}; \Theta_{m-1})} L(f_{m-1}; \Theta_{m-1}),$$

it can be challenging to compute it using automatic differentiation (AD) due to the fact that $L(f_{m-1}; \theta)$ is typically a leaf node in the computational graph.

Fortunately, we can still utilize the formulation in equation 8 to establish an appropriate GB algorithm for PINNs. Rather than computing the gradient as previously mentioned, we introduce the core algorithm of this paper. Instead of determining $h_m$ as depicted in equation 6, a neural network is employed to represent $h_m$, and it is trained following the same procedure as PINNs. Specifically, once the training of the $(m-1)$-th step concludes, we incorporate an additional, pre-selected neural network, denoted as $\rho_m h_m(\boldsymbol{x}; \theta_m)$, to the preceding predictor $f_{m-1}(\boldsymbol{x}; \Theta_{m-1})$. Subsequently, a fresh batch of training points is generated to facilitate

the training process. This iterative procedure allows us to gradually minimize the loss and improve the accuracy of the predicted solution.

It is important to note that the neural networks utilized in the proposed GB PINNs algorithm do not need to possess a consistent structure. In fact, they can be composed of a variety of surrogate models, as long as they have the same input and output dimensions. Examples of such models include MLPs, Fourier feature networks, radial basis neural networks, and even finite element representations. This flexibility allows for a more versatile approach to minimizing the loss and improving the accuracy of the approximation to the exact latent solution.

Another notable point regarding our proposed method is its foundation in GB from conventional machine learning tasks. However, GB PINNs is not merely a straightforward amalgamation of GB and PINNs. This is primarily due to the distinct nature of GB in this context. The most significant distinction lies in how we determine the model update, $h_m$. While traditional GB derives the update from the gradient in function space, as indicated in equation 6, our proposed GB PINNs, albeit inspired by conventional GB, derives the update using a neural network, following the PINNs training procedure. The common ground between these two methodologies is represented by equation 8, yet the approach to the model update $h_m$ diverges in each case.

### 3.2   Determination of the learning rate parameter, $\rho_m$

In concluding the algorithm description, we must address the determination of the parameter, $\rho_m$. In GB training, the term $\rho_m$ from equation 5 acts as a learning rate within the function space gradient descent and additionally modulates the magnitude of $h_m(\boldsymbol{x}; \theta_m)$. Since the learning rate is inherently a hyperparameter in machine learning, a universal method to obtain an optimal value often remains elusive. Yet, we can derive insights from the intuitive and empirical aspects of the training process.

In many PINNs applications, upon the completion of the training for $f_{m-1}(\boldsymbol{x}; \Theta_{m-1})$ with a notably low loss value, a reliable predictor emerges. This suggests that the relative error between the current predictor and the actual ground truth (e.g., the relative $l^2$ error defined later) is minimal. Even when the loss is not minimal, the neural network's output typically surpasses its random initial state. In other words, even if post-training accuracy falls short, it's evident that the neural network has absorbed some features during the training phase. Subsequently, the additive model, $h_m(\boldsymbol{x}; \theta_m)$, is trained to further minimize the error, leading us to expect a decreasing magnitude of $h_m(\boldsymbol{x}; \theta_m)$ with each iteration step, $m$. By extension, it's logical to expect a corresponding decrement in $\rho_m$ over $m$.

In the subsequent experiments, we adopt an exponential decay for $\rho_m$, aligning it with conventional gradient descent approaches. Nevertheless, despite the aforementioned rationale, it's imperative to underscore that this configuration is empirical and might not universally apply.

### 3.3   Comprehensive Overview of the GB PINNs Algorithm

The proposed algorithm can be summarized as follows:

The proposed algorithm, described in Algorithm 1, utilizes a sequence of neural networks and an iterative update procedure to minimize the loss gradually. At each iteration step $i$, the forward prediction relies on the union parameter set $\Theta_i$, while the backward gradient propagation is only performed on $\theta_i$. This results in a mild increase in computational cost during the training of GB iteration. The simplicity of this algorithm allows practitioners to easily transfer their PINNs codes to GB PINNs'. In the following section, we will demonstrate that this simple technique can enable PINNs to solve many problems that were previously intractable using the original formulation of Raissi et al. (2019).

Additionally, the proposed GB PINNs algorithm also introduces another dimension of flexibility in terms of network architecture design, namely the combination of different neural networks. This opens up new opportunities for fine-tuning the architecture to minimize PDE residual losses and improve overall predictive accuracy. As shown in the following section, the performance of GB PINNs is relatively insensitive to the specific choice and arrangement of networks, as long as their capacity is sufficient.

---

**Algorithm 1** Gradient boosting physics informed neural network.

**Input:**

 A baseline neural network $f_0(\boldsymbol{x}; \theta_0)$ and an ordered neural network set $\{h_m(\boldsymbol{x}; \theta_m)\}_{m=1}^M$ that contains models going to be trained in sequence;

 A set of learning rate $\{\rho_m\}_{m=0}^M$ that correspond to $\{f_0(\boldsymbol{x}; \theta_0)\} \cup \{h_m(\boldsymbol{x}; \theta_m)\}_{m=1}^M$. Usually, $\rho_0 = 1$ and $\rho_m$ is decreasing in $m$;

 Set $f_m(\boldsymbol{x}; \Theta_m) = f_{m-1}(\boldsymbol{x}; \Theta_{m-1}) + \rho_m h_m(\boldsymbol{x}; \theta_m)$, for $m = 1, 2, 3, \cdots, M$.

 Given PDEs problem 1-2, establish the corresponding loss 3.

**Output:**

1: Train $f_0(\boldsymbol{x}; \theta_0) = \rho_0 f_0(\boldsymbol{x}; \theta_0)$ to minimize loss 3.

2: **for** $m = 1$ to M **do**

3:  In $f_m(\boldsymbol{x}; \Theta_m) = f_{m-1}(\boldsymbol{x}; \Theta_{m-1}) + \rho_m h_m(\boldsymbol{x}; \theta_m)$, set trainable parameters as $\theta_m$. Train $f_m(\boldsymbol{x}; \Theta_m)$ to minimize loss 3.

4: **end for**

5: **return** $f_M(\boldsymbol{x}; \Theta_M)$ as a predictor of the solution of 1-2 for any point in $\overline{\Omega}$.

---

### 3.4 Conceptual insights of GB PINNs

In this subsection, we elucidate why GB PINNs can be expected to outperform vanilla PINNs, considering two primary reasons.

Firstly, GB, a recognized technique in traditional machine learning tasks like tree models, systematically adds new models in each step to rectify its predecessor's errors. This principle extends to GB PINNs as well. Although the update model $h_m$ for each step is derived from a new network rather than gradient descent in function space, every update reduces the residuals left by the preceding model. As more models are integrated, the entire system's complexity amplifies, enabling it to encapsulate more intricate patterns in PDE solutions. This behavior is evident in our first three numerical examples concerning sharp gradient solutions.

Secondly, as emphasized in Wang et al. (2022), the neural tangent kernel (NTK) of expansive and deep neural networks tends to remain consistent during training, potentially hindering the learning of solutions with sharp gradients. It's also observed that large neural networks face the challenge of 'lazy training' Chizat et al. (2019). Here, the network's weights rarely exhibit significant shifts during training, which constrains its expressive capacity. While shrinking the network's size can alleviate these issues, it risks underfitting if excessively reduced. GB PINNs, through their algorithmic design, harness smaller networks to counteract the lazy training and NTK stability. They subsequently leverage larger networks to address potential underfitting.

Such insights also inspire an approach where a compact network forms the initial solution approximation, which is successively refined using heftier networks. [1]

In the next section, we will delve into numerical experiments to validate our proposed approach. Although we have not set strict prerequisites for our method's application, its efficacy for sharp gradient and certain nonlinear time-evolution problems with periodic boundary conditions will be demonstrated. Nonetheless, the method isn't universally optimal, showing limitations with complex challenges like conservation laws, which we will touch upon in the conclusion.

## 4 Numerical Experiments

In this section, we will demonstrate the effectiveness of the proposed GB PINNs algorithm through a comprehensive set of numerical experiments. To simplify the notation, we use a tuple of numbers to denote the neural network architecture, where the tuple represents the depth and width of the layers. For example, a neural network with a two-dimensional input and a one-dimensional output, as well as two hidden layers

---

[1]It's important to clarify that our proposed neural network structure remains conceptual and might not always deliver peak results. Depending on the distinct problem, the selection and organization of neural networks could differ. Our numerical experiments will offer deeper perspectives on tailoring neural networks to specific challenges.

Table 1: Default Experiment Set up

| Name | Value |
|---|---|
| Activation function | GeLU |
| Method to initialize the neural network | Xavier |
| Optimizer | Adam |
| learning rate | $10^{-3}$ |
| learning rate decay period | $10,000$ |
| learning rate decay rate | $0.95$ |

with width 100 is represented as $(2, 100, 100, 1)$. Our default experimental setup is summarized in Table 1, and will be used in all experiments unless otherwise specified.

To quantify the model's accuracy, we use the relative $l^2$ error over a set of points $\{x_i\}_{i=1}^N$:

$$\text{Error} = \frac{\sum_{i=1}^N |u_{pred}(x_i) - u_{true}(x_i)|^2}{\sum_{i=1}^N |u_{true}(x_i)|^2}.$$

In our analysis, we assess the error across a defined set of grid points. In the subsequent experiments, we produce a set of $1,000$ equidistant points for each dimension within the domain. These points are then combined using the Cartesian product to establish the grid coordinates.

### 4.1 1D singular perturbation

In this first example, we utilize GB PINNs to solve the following 1D singular perturbation problem.

$$-\varepsilon^2 u''(x) + u(x) = 1, \qquad \text{for } x \in (0, 1),$$
$$u(0) = u(1) = 0.$$

The perturbation parameter, $0 < \varepsilon \ll 1$, is set to $10^{-4}$ in this case. The exact solution to this problem is given by

$$u(x) = 1 - \frac{e^{-x/\varepsilon} + e^{(x-1)/\varepsilon}}{1 + e^{-1/\varepsilon}}.$$

Despite the boundedness of the solution, it develops boundary layers at $x = 0$ and $x = 1$ for small values of $\varepsilon$, a scenario in which traditional PINNs have been known to perform poorly.

Utilizing the notation established in Algorithm 1, we designate $f_0$ as $(1, 50, 1)$, $h_1$ as $(1, 100, 1)$, $h_2$ as $(1, 100, 100, 1)$, $h_3$ as $(1, 100, 100, 100, 1)$ and $h_4$ as a Fourier feature neural network $(1, 50, 50, 1)$ with frequencies ranging from 1 to 10. For the sake of clarity in subsequent examples, we will simply present a series of network architectures, which will implicitly represent the $f_0$ and $h_m$ $(m = 0, 1, 2, \cdots)$ configurations in sequence. The details of the Fourier feature method used in this study can be found in the appendix A.1. The step size $\rho_m$ in equation 5 was set to $0.5^m$, where $m = 0, 1, \cdots, 4$ is the model index.

For each GB iteration, we train $10,000$ steps using a dataset of $10,000$ uniform random points in $(0, 1)$. The weights in the loss function 3 are set to $\omega_e = 1$ and $\omega_b = 10$, respectively, and the batch size for PDE residual computation is $10,000$.

The output of GB PINNs is shown in Figure 1, where the relative $l^2$ error is found to be 0.43%. The boundary layers at $x = 0$ and $x = 1$ are clearly visible in the solution, which is a result of the thinness of the layers and the almost right angle curvature of the solution at these points. Despite the singularity present in the solution, GB PINNs were able to provide an accurate solution for this problem. To further highlight the contribution of GB PINNs in this example, an ablation study was conducted. A vanilla PINNs approach, using a network structure of $(1, 256, 256, 256, 256, 1)$ and $50,000$ training steps, was used to solve the same problem. To make this comparison fair, we use the same amount of training points in total. Therefore, we set the batch size of the training points as the same as before. The training takes 249.47s. Notably, even though

this network possesses greater depth and width than any single network in the GB PINNs ensemble, the resulting relative $l^2$ error is a much higher 12.56%, as shown in Figure 2. Additional experiments including ablation studies and comparisons can be found in appendix 2.

However, getting this high level of accuracy comes with its costs. Using a series of neural networks instead of just one means more time and memory are needed. The training times for the GB PINNs networks are 57.44s, 77.63s, 126.96s, 196.08s, and 259.44s, respectively. The peak memory requisition touched 0.28GB. In a scenario where only the largest configuration within the GB PINNs' network spectrum, specifically $(1, 100, 100, 100, 1)$, is utilized, the memory footprint scales down to 0.16GB. Compared to the standard PINNs, this method takes about three times longer and uses twice the memory. But the accuracy was more than 20 times better. Another discernible trend is the incremental rise in training time in correlation to the network's order. As we advance to training the $m$-th network, even with the parameters of the preceding networks remaining static, the computational intensity during both forward and backward propagations escalates, leading to protracted training durations.

Furthermore, we demonstrate the robustness of our algorithm against the choice of network structure and arrangement through this example. We solve the problem using a variety of networks, including $(1, 50, 1)$, $(1, 100, 1)$, $(1, 100, 100, 1)$, followed by a Fourier feature network $(1, 100, 100, 100, 1)$ with frequencies ranging from 1 to 10. The resulting relative $l^2$ error is around 1.1%, which is comparable to the previously mentioned result. It is possible that there exists a specific set of hyperparameters and configurations that would allow a single neural network to perfectly solve this problem. After all, by the universal approximation theorem Hornik et al. (1989), even a neural network with a simple structure possesses the ability to approximate a complicated function. However, the fine-tuning of hyperparameters is a common challenge in machine learning tasks and can consume significant computational resources. Contrarily, GB PINNs mitigate the intricacies associated with adjusting network structures, as the multiple networks intrinsically refine the outputs, saving both effort and computational resources. It is worth noting, however, that this does not obviate the need to fine-tune other hyperparameters, such as the learning rate.

An additional observation from our ablation study and further detailed in Appendix A.2, highlights a limitation inherent to the proposed GB PINNs. As illustrated in the final row of Table 2, enriching the neural network spectrum by adding more networks failed to improve accuracy. A closer examination of Figure 1 offers some insights into this phenomenon. The current neural network configuration generates a solution where errors predominantly localize at both endpoints. This presents a challenge for neural networks, as the error distribution lacks smoothness. Consequently, simply augmenting the number of networks is unlikely to mitigate this issue.

In conclusion, we juxtapose our GB PINNs framework with a rudimentary ensemble approach for comparison. Employing the same architecture outlined initially—$(1, 50, 1)$, $(1, 100, 1)$, $(1, 100, 100, 1)$, $(1, 100, 100, 100, 1)$, and a Fourier feature neural network $(1, 50, 50, 1)$ with frequency ranges from 1 to 10—we assign equal weightings of 0.2 to each model. The cumulative output is consequently the arithmetic mean of these five individual networks. All models are concurrently trained using the Adam optimizer over a span of $50,000$ steps. The findings are illustrated in Figure 3.

While the ensemble method attains a relatively low error rate of 0.89%, a closer examination reveals a notable error in the boundary layer, as is evident from the figure. Although the maximal error is nearly commensurate with that of our GB PINNs, the ensemble approach exhibits pronounced jumps at the boundary layer, thereby distorting the overall solution.

## 4.2   2D singular perturbation with boundary layers

In this example, we aim to solve the Eriksson-Johnson problem, which is a 2D convection-dominated diffusion equation. As previously noted in the literature, such as in Demkowicz & Heuer (2013), this problem necessitates the use of specialized finite element techniques in order to obtain accurate solutions, such as the Discontinuous Petrov Galerkin (DPG) finite element method.

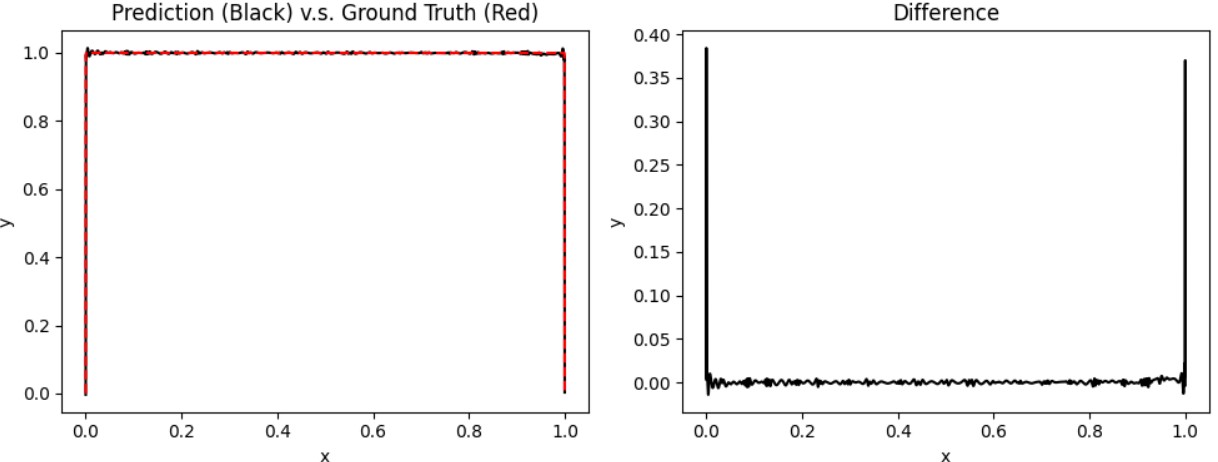

Figure 1: Prediction of singular perturbation problem by GB PINNs, $\varepsilon = 10^{-4}$. Left: predicted solution (black) v.s. ground truth (red). Right: pointwise error.

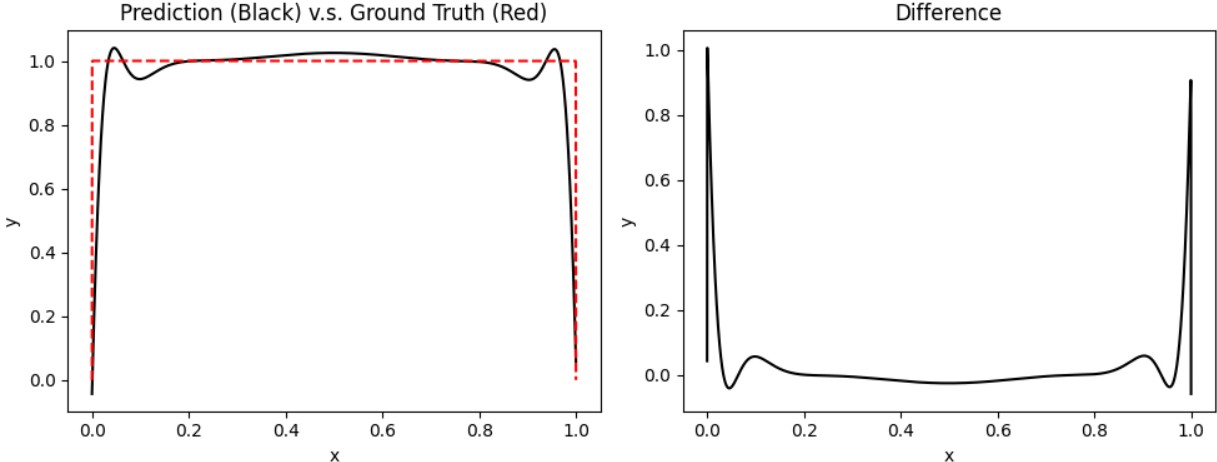

Figure 2: Prediction of singular perturbation problem by PINNs for ablation study, $\varepsilon = 10^{-4}$. Left: predicted solution (black) v.s. ground truth (red). Right: pointwise error.

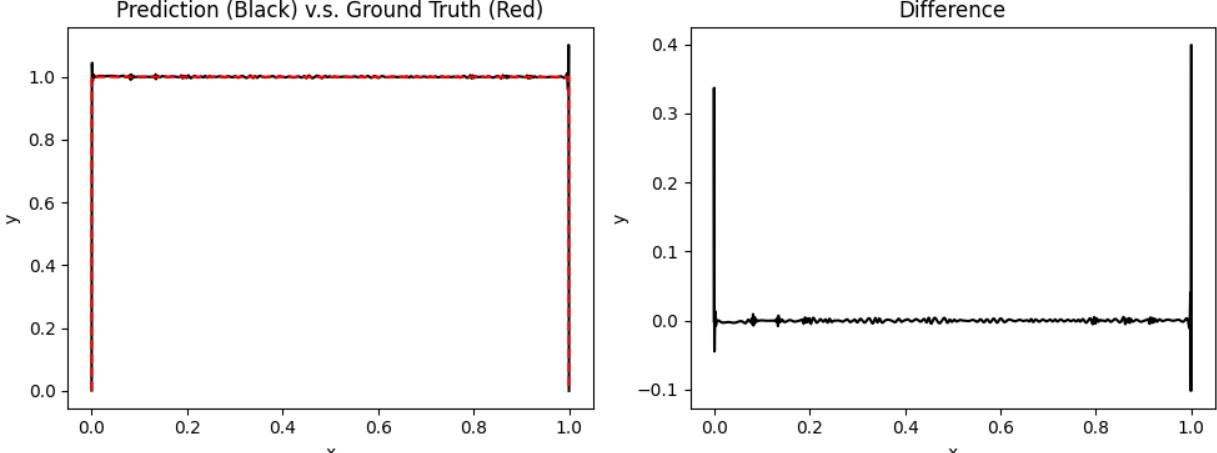

Figure 3: Rudimentary ensemble method by taking the mean of the five networks, $\varepsilon = 10^{-4}$. Left: predicted solution (black) v.s. ground truth (red). Right: pointwise error.

Let $\Omega = (0,1)^2$. The model problem is

$$-\varepsilon \Delta u + \frac{\partial u}{\partial x} = 0 \qquad \text{in } \Omega,$$
$$u = u_0 \qquad \text{on } \partial \Omega.$$

The manufactured solution is

$$u(x,y) = \frac{e^{r_1(x-1)} - e^{r_2(x-1)}}{e^{-r_1} - e^{-r_2}} \sin(\pi y) \quad \text{with} \quad r_{1,2} = \frac{-1 \pm \sqrt{1 + 4\varepsilon^2\pi^2}}{-2\varepsilon}.$$

In this example, we set $\varepsilon = 10^{-3}$. To resolve this problem, we sequentially employ a range of neural network architectures as follows: $(2, 50, 1)$, $(2, 100, 1)$, $(2, 100, 100, 1)$, $(2, 100, 100, 100, 1)$, $(2, 100, 100, 1)$, culminating in a Fourier feature network $(1, 100, 100, 1)$ with frequencies ranging from 1 to 50. For each iteration of our GB algorithm, we train for $20,000$ steps. We set the weights in equation 3 as $\omega_e = 1$ and $\omega_b = 10,000$, respectively. The batch sizes for PDE residuals and boundaries are set at $10,000$ and $50,00$, respectively. The predicted solution is visualized in Figure 4. We can see that our model prediction is in good agreement with the ground truth, with a relative $l^2$ error of 1.03%. The training times of each individual network are 92.72s, 119.62s, 204.66s, 329.35s, and 459.93s, respectively. The maximum memory consumption reached 0.71GB. However, when only using the largest network configuration, $(2, 100, 100, 100, 1)$, the peak memory usage stands at 0.31GB.

Notably, there is a boundary layer present on the right side of the boundary $(x = 1)$, which is not easily recognizable to the naked eye due to its thinness. However, GB PINNs are able to provide a reasonable degree of predictive accuracy even in this challenging scenario.

To further demonstrate the efficacy of our proposed method, we also attempted to solve this problem using a single fully connected neural network of architecture $(2, 256, 256, 256, 256, 1)$. We train this network by $20,000$ steps under the same hyperparameter settings as before. For a fair comparison, we maintain the same total number of training points. Consequently, we have increased the batch size of the training points by fivefold. However, the resulting relative $l^2$ error was 57.66%. This training consumed 811.82s of time. As can be seen in Figure 5, there is a significant discrepancy between the predicted solution and the reference solution. Additional experimental results, including an ablation study and comparisons, can be found in Appendix 3.

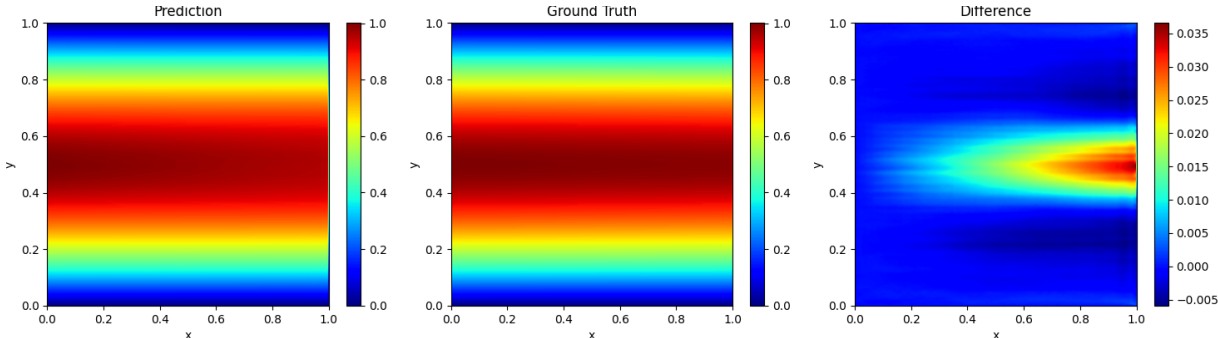

Figure 4: Prediction of 2D singular perturbation with boundary problem by GB PINNs, $\varepsilon = 10^{-3}$. Left: predicted solution. Middle: ground truth. Right: pointwise error.

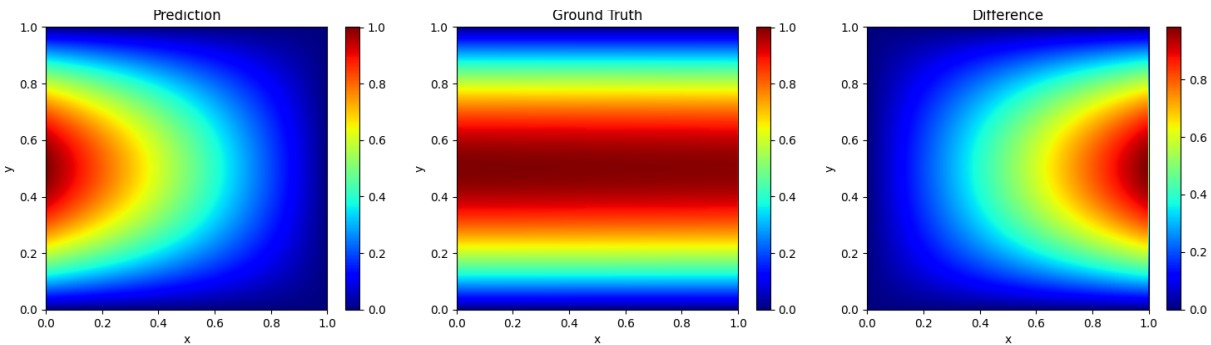

Figure 5: Prediction of 2D singular perturbation with boundary problem by PINNs, $\varepsilon = 10^{-3}$. Left: predicted solution. Middle: ground truth. Right: pointwise error.

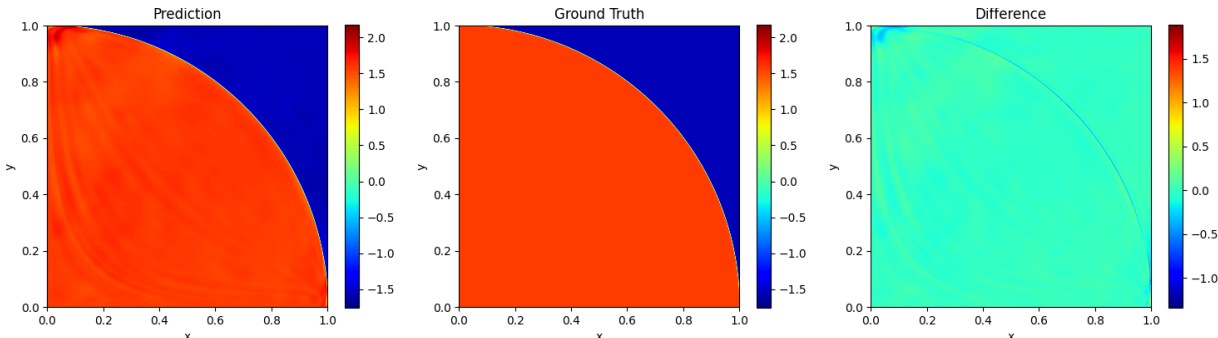

Figure 6: Prediction of 2D singular perturbation with interior boundary problem by GB PINNs, $\varepsilon = 10^{-4}$. Left: predicted solution. Middle: ground truth. Right: pointwise error.

### 4.3  2D singular perturbation with an interior boundary layer

In this example, we address a 2D convection-dominated diffusion problem featuring curved streamlines and an interior boundary layer. The model problem is

$$-\varepsilon \Delta u + \beta \cdot \nabla u = f \qquad \text{in } \Omega,$$
$$u = u_0 \qquad \text{on } \partial\Omega,$$

with $\beta = e^x(\sin(y), \cos(y))$ and $f$, $u_0$ are defined such that

$$u(x, y) = \arctan\left(\frac{1 - \sqrt{x^2 + y^2}}{\varepsilon}\right).$$

This example has been solved by DPG finite element method in Demkowicz & Heuer (2013). A specific value of $\epsilon = 10^{-4}$ was chosen for the purpose of this study. The neural network architectures are sequentially employed in the following order: $(2, 200, 200, 200, 1)$, $(2, 100, 100, 100, 1)$, $(2, 100, 100, 1)$, culminating in a Fourier feature network $(2, 50, 50, 1)$ with frequency ranging from 1 to 5. The weights for the loss function in equation 3 were set as $\omega_e = 1$ and $\omega_b = 10,000$, respectively. The batch size for the PDE residual and boundary were set to $10,000$ and $5,000$, respectively. For each iteration of our GB algorithm, we train for $20,000$ steps. The results of this study are shown in Figure 6 and exhibit a relative $l^2$ error of 3.37%. The training times for each individual network are 409.65s, 463.84s, 525.87s, and 562.61s, respectively. The peak memory allocation for the GB PINNs and the largest configuration are 0.84GB and 0.63GB, respectively.

As a part of an ablation study, we resolve this problem using a fully connected neural network architecture of $(2, 512, 512, 512, 512, 1)$, while maintaining the other configurations as same as the previous experiment. We train this network by $20,000$ steps. For a fair comparison, we maintain the same total number of training points. Consequently, we have increased the batch size of the training points by fourfold. This training consumed 1067.96s of time. The relative $l^2$ error obtained in this case is 43%. The predictions and the corresponding errors are depicted in Figure 7. Additional experiments pertaining to the ablation study and comparisons can be found in the appendix, section 4.

### 4.4  2D nonlinear reaction-diffusion equation

In this example, we investigate the solution of a time-dependent nonlinear reaction-diffusion equation. As demonstrated in Krishnapriyan et al. (2021), conventional PINNs have been shown to be inadequate in accurately learning the solution of such equations.

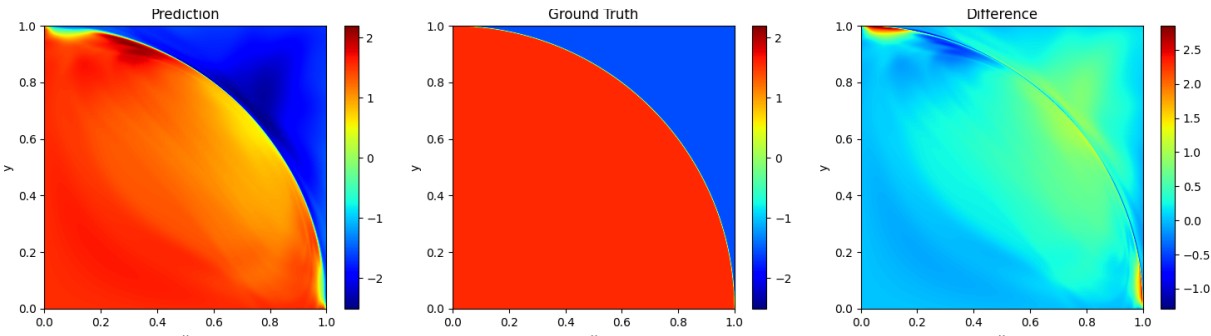

Figure 7: Prediction of 2D singular perturbation with interior boundary problem by PINNs, $\varepsilon = 10^{-4}$. Left: predicted solution. Middle: ground truth. Right: pointwise error.

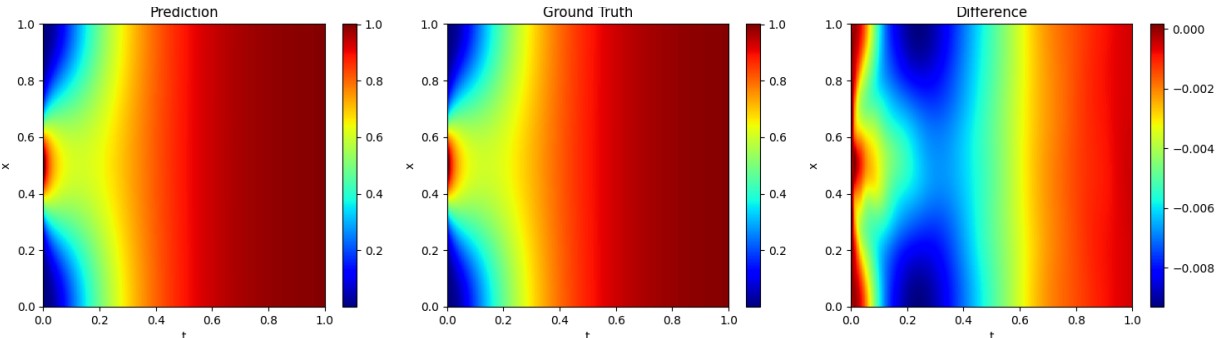

Figure 8: Prediction of nonlinear reaction-diffusion equation by GB PINNs. Left: predicted solution. Middle: ground truth. Right: pointwise error.

Let $\Omega = (0, 2\pi)$. The model problem is

$$\frac{\partial u}{\partial t} - 10\frac{\partial^2 u}{\partial x^2} - 6u(1-u) = 0, \quad x \in \Omega, t \in (0, 1],$$

$$u(x, 0) = h(x) \quad x \in \Omega,$$

with periodic boundary conditions, where

$$h(x) = e^{-\frac{(x-\pi)^2}{2(\pi/4)^2}}.$$

In order to impose an exact periodic boundary condition, we use $(\sin(x), \cos(x))$ as the spatial input instead of $x$, while maintaining the temporal input unchanged. This eliminates the need for boundary loss. Additionally, we include an additional loss term for the initial condition in the loss function, equation 3. For this problem, we employ neural network architectures in the following sequential order: $(2, 200, 200, 200, 1)$, $(2, 100, 100, 100, 1)$, $(2, 100, 100, 1)$. The weights for the PDE residual and initial condition loss are set to 1 and $1,000$, respectively. The batch sizes for the PDE residual and initial condition loss are $20,000$ and $1,000$, respectively. For each iteration of our GB algorithm, we train for $20,000$ steps. We present our results in Figure 8. The relative $l^2$ error is 0.58%. As shown in Krishnapriyan et al. (2021), the relative error for traditional PINNs with $\rho = \nu = 5$ is 50%. A comparison between the exact solution and the PINNs' prediction can also be found in the figure. The training times for each individual network are 391.99s, 483.21s, and 579.77s, respectively. The peak memory allocation for the GB PINNs and the largest configuration are 0.99GB and 0.77GB, respectively.

In the aforementioned study, Krishnapriyan et al. (2021) proposed a sequence-to-sequence learning approach to address this problem, achieving a relative $l^2$ error of 2.36% for $\rho = \nu = 5$. This approach begins by

uniformly discretizing the temporal domain, resulting in sequential subintervals. Each of these temporal subintervals is then combined with the spatial domain, forming distinct subdomains. The problem is solved sequentially by applying traditional PINNs through these spatio-temporal subdomains. For the cited example above, the sequence-to-sequence model trained the neural network across 20 such intervals, necessitating the solution of 20 consecutive problems via PINNs.

In contrast, our methodology, executed with $\nu = 10$ and $\rho = 6$, which is a notably more complex setting, required training for only three networks. Rather than partitioning the domain at the physical level and employing multiple learners to construct solutions on subdomains before amalgamating them, our strategy utilizes supplemental learners to reinforce the base learner, thereby enhancing its precision. Remarkably, our approach yielded an error rate nearly four times lower than the former method, signifying a substantial enhancement.

## 5 Conclusion

In this paper, we propose a GB PINNs algorithm, which utilizes multiple neural networks in sequence to predict solutions of PDEs. The algorithm is straightforward to implement and does not require extensive fine-tuning of neural network architectures. Additionally, the method is flexible and can be easily integrated with other PINNs techniques. Our experimental results demonstrate its effectiveness in solving a wide range of intractable PDE problems.

However, it should be noted that the algorithm has some limitations. Firstly, it is not suitable for solving conservation laws with derivative blow-ups, such as the inviscid Burgers' equation and the Sod shock tube problem. This is due to the lack of sensitivity of these equations' solutions to PDE loss. The addition of more neural networks alone cannot overcome this issue. Secondly, the optimal combination of neural networks is not always clear, and the current experimental selection is mostly based on experience and prior estimation of the PDE problem. Further research into the theoretical and quantitative analysis of this method is an interesting direction for future work.

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

# A Appendix

## A.1 Fourier features networks

In this subsection, we present the Fourier feature network structure that we employed in our experiments. For a more in-depth explanation of Fourier features, please refer to Tancik et al. (2020).

Given an input $\boldsymbol{x} \in \mathbb{R}^{N \times d}$ for the neural network, where $N$ is the batch size and $d$ is the number of features, we encode it as $\boldsymbol{v} = \gamma(\boldsymbol{x}) = [\cos(2\pi\boldsymbol{x}\boldsymbol{B}), \sin(2\pi\boldsymbol{x}\boldsymbol{B})] \in \mathbb{R}^{N \times 2m}$, where $\boldsymbol{B} \in \mathbb{R}^{d \times m}$, and $m$ is half of the output dimension of this layer. The encoded input $\boldsymbol{v}$ is then passed as input to the subsequent hidden layers, while the rest of the neural network architecture remains the same as in a traditional MLP. Different choices of $\boldsymbol{B}$ result in different types of Fourier features.

In our experiment, we utilized the axis Fourier feature. We first selected a range of integers as our frequencies and denoted them as $\boldsymbol{f} = (f_0, f_1, \cdots, f_p) \in \mathbb{R}^p$. We then constructed a block matrix $\boldsymbol{B} = [\boldsymbol{F}_0, \boldsymbol{F}_1, \cdots, \boldsymbol{F}_p] \in \mathbb{R}^{d \times dp}$, where $\boldsymbol{F}_i$ is a diagonal matrix with all of its diagonal elements set to $f_i$. This results in the desired matrix $\boldsymbol{B} \in \mathbb{R}^{d \times dp}$, and $dp = m$ is half of the output dimension for this layer.

## A.2 Additional ablation study results

In this subsection, we present additional experiments to demonstrate the relative $l^2$ error for various neural network selections. The objective of these experiments is to demonstrate the following:

1. The use of GB PINNs can significantly improve the accuracy of a single neural network.

2. The method is relatively insensitive to the selection of neural networks, as long as the spectral and capacity are sufficient.

In order to simplify the presentation, we will only display the hidden layers of fully-connected neural networks and utilize the list notation in Python. For instance, $[50] * 3$ represents three hidden layers, each containing 50 neurons. The input and output layers are implied by context and not explicitly shown. For Fourier feature neural networks, we denote them using the notation $F_k$, where $k$ represents the range of frequencies used (e.g., $F_{10}[100] * 2$ denotes a Fourier feature neural network with frequencies ranging from 1 to 10, and two hidden layers with 100 neurons each). In the case of the 2D nonlinear reaction-diffusion equation problem, we use the prefix $P$ to indicate the use of a periodic fully-connected neural network. The weight for each study is set to $2^{-n}$, where $n$ is the index of the neural network.

In order to facilitate clear comparisons, the reported results will be presented in the last line of the following tables.

In the following results, it becomes evident that incorporating a Fourier feature network enhances performance, primarily by smoothing the neural tangent kernels. However, we argue that relying solely on a single Fourier feature network is insufficient for attaining the level of accuracy achieved by GB PINNs. As corroborated by data presented in Table 2 through Table 4, specifically in the penultimate row, there exists a discernible gap in accuracy between a standalone Fourier feature network and GB PINNs. This discrepancy is particularly noticeable in 2D problems. Therefore, the superior accuracy of GB PINNs can be attributed to the synergistic interaction among multiple networks, rather than the implementation of a Fourier feature network alone.

Table 2: Ablation study for 1D singular perturbation

| Neural network structures | Relative $l^2$ error |
|---|---|
| $[50], [100]$ | 31.36% |
| $[50], [100], [100] * 2$ | 11.05% |
| $[50], [100], [100] * 2, [100] * 3$ | 10.79% |
| $[50], [100], [100] * 2, [100] * 3, [100] * 2, F_{10}[50] * 2$ | 0.85% |
| $[50], [100], [100] * 2, F_{10}[100] * 3$ | 1.1% |
| $[512] * 6$ | 15.16% |
| $F_{10}[256] * 4$ | 0.60% |
| $F_{10}[50] * 2$ | 1.27% |
| $[50], [100], [100] * 2, [100] * 3, F_{10}[50] * 2$ | 0.43% |

Table 3: Ablation study for 2D singular perturbation with boundary layers

| Neural network structures | Relative $l^2$ error |
|---|---|
| $[50], [100]$ | 61.56% |
| $[50], [100], [100] * 2$ | 57.01% |
| $[50], [100], [100] * 2, [100] * 3$ | 57.25% |
| $[50], [100], [100] * 2, [100] * 3, [100] * 2$ | 57.23% |
| $[50], [100], [100] * 2, F_{50}[100] * 2$ | 2.67% |
| $F_{50}[256] * 4$ | 4.01% |
| $F_{50}[100] * 2$ | 15.56% |
| $[50], [100], [100] * 2, [100] * 3, [100] * 2, F_{50}[100] * 2$ | 1.03% |

Table 4: Ablation study for 2D singular perturbation with an interior boundary layer

| Neural network structures | Relative $l^2$ error |
|---|---|
| $[200] * 3, [100] * 3, [100] * 2$ | 8.69% |
| $[200] * 3, [100] * 3, F_5[50] * 2$ | 6.46% |
| $[200] * 3, [100] * 3, [100] * 2, F_5[50] * 2, [100] * 3$ | 6.03% |
| $F_5[512] * 4$ | 38.9% |
| $F_5[50] * 2$ | 16.15% |
| $[200] * 3, [100] * 3, [100] * 2, F_5[50] * 2$ | 3.37% |

Table 5: Ablation study for 2D nonlinear reaction-diffusion equation

| Neural network structures | Relative $l^2$ error |
|---|---|
| $P[200] * 3$ | 1.18% |
| $P[200] * 3, P[100] * 3$ | 1.18% |
| $P[200] * 3, P[100] * 3, P[100] * 2, P[100] * 3$ | 0.59% |
| $P[200] * 3, P[100] * 3, P[100] * 2$ | 0.58% |

