# OpenReview forum: "Ensemble learning for Physics Informed Neural Networks: a Gradient Boosting approach"
_TMLR — Rejected by TMLR_

### Review · Reviewer_qiNs · 2023-08-07

**Summary Of Contributions:**

The present paper advocates for the training of PINNs using a boosting method referred to as GB PINNs. The authors define a sequential procedure of building an additive predictor successively adding neural networks, typically starting with small models and using larger models subsequently. The parameters of each new neural network are trained by stochastic gradient descent on the usual loss of PINNs combining PDE and boundary residuals. The authors present results on 4 different PDEs on which PINNs are known to have difficulties and find some significant improvements when doing sequential learning.

**Audience:**

Yes

**Claims And Evidence:**

No

**Requested Changes:**

Critical change:
- Demonstrate the claim of robustness to hyper-parameters or modify the writing.

Suggested improvements: Please refer to weaknesses above.

Additionally, could the authors clarify the following points:
- Bottom of page 5: How are the points chosen to quantify the error? Do they coincide with points used in the training loss or not?
- When retraining a newly added component of the GB PINN, are the same training points used? If new training points are sampled, it seems that a fair comparison with other methods should include as many training points as the total number of training points seen by GB PINN method.
- What is the difference between the method proposed in this paper and the sequence-to-sequence learning of Krishnapriyan et. al. ?

Minor comments:
- There are some repetitions when citing papers or equations (e.g. “equation equation [8])
- page 4, The acronym “AD” which probably refers to Automatic Differentiation is not introduced before being used I believe.

**Strengths And Weaknesses:**

Strengths:
- Several types of PDEs are considered and significant improvement over single networks PINNs are reported.
- The description of the context is satisfactory.

Weaknesses:
- The analysis of the experiments could advantageously be extended. While ablation studies are proposed in the appendix, it would be interesting to investigate more extensively whether larger/more trained single network PINNs can close the gap in performance to the proposed GB PINNs. For instance, for the experiment presented in 4.3 the single network the method is compared to is the same as the initial network of the respective GB PINNs.
- There is no clear discussion of the computational costs in the proposed experiments. When reporting a better accuracy than a single network PINNs, it would be useful to understand a higher computational budget was necessary.
- The claim that the method is more robust to hyper-parameters tuning is not strongly supported by the reported data. The claim would require reporting results for different architecture and learning rates for instance for GB PINNs and competitors.

Provided this last observation, I cannot confirm that the claims are supported by clear evidence. However, it should be easy for the authors either to report the experiments they conducted to get to this conclusion or to modify the writing to avoid any confusion.

---

> ### Author Response · Authors · 2023-08-15
> **Modification on the first version**
>
> Dear Reviewer,
>
> Thank you for your thoughtful feedback on our manuscript. We have made necessary modifications in response to your comments. For your convenience, we have highlighted these changes in red in the revised manuscript. Below, we provide specific responses to each of your points, and we trust these address your concerns adequately.
>
> 1. *The analysis of the experiments could advantageously be extended. While ablation studies are proposed in the appendix, it would be interesting to investigate more extensively whether larger/more trained single network PINNs can close the gap in performance to the proposed GB PINNs. For instance, for the experiment presented in 4.3 the single network the method is compared to is the same as the initial network of the respective GB PINNs.*
>
> **[Response] We concur with your suggestion. Accordingly, we've adjusted the reference vanilla PINNs’ structure in numerical experiments 4.1-4.3 to incorporate a larger neural network. These are typically more extensive than the largest network in the GB PINNs’ network spectrum.**
>
> 2. *There is no clear discussion of the computational costs in the proposed experiments. When reporting a better accuracy than a single network PINNs, it would be useful to understand a higher computational budget was necessary.*
>
> **[Response] We appreciate your insight. We have now incorporated data regarding training time and peak memory allocation in the numerical experiments. Specifically, we present the training time for each individual network and for the vanilla PINNs. Regarding memory, we've detailed the peak memory allocation for GB PINNs and for scenarios where only the largest network in the GB PINNs network set is utilized.**
>
> 3. *The claim that the method is more robust to hyper-parameters tuning is not strongly supported by the reported data. The claim would require reporting results for different architecture and learning rates for instance for GB PINNs and competitors.*
>
> **[Response] Thank you for highlighting this. Our intention was to emphasize that, while users may invest significant effort fine-tuning the network structure, they still need to adjust hyperparameters like the learning rate. We have clarified this in the manuscript, particularly in the lines preceding section 4.2 on page 7.**
>
> 4. *How are the points chosen to quantify the error? Do they coincide with points used in the training loss or not?*
>
> **[Response] We utilize points different from the training set to evaluate the loss. We've added a clarification before section 4.1 on page 6.**
>
> 5. *When retraining a newly added component of the GB PINN, are the same training points used? If new training points are sampled, it seems that a fair comparison with other methods should include as many training points as the total number of training points seen by GB PINN method.*
>
> **[Response] Indeed, in PINNs-related methods, including GB PINNs, we typically generate a new set of training points for each component network rather than reusing them. You've raised a valid point regarding fair comparisons. To address this, we've modified our comparative experiments in sections 4.1 to 4.3 between GB PINNs and vanilla PINNs. We've increased the batch size for the vanilla PINNs to ensure that both methods are exposed to the same number of training points.**
>
> 6. *What is the difference between the method proposed in this paper and the sequence-to-sequence learning of Krishnapriyan et. al. ?*
>
> **[Response] We've introduced a brief overview of the sequence-to-sequence method and its comparison with GB PINNs at the end of page 11.**
>
> 7. *There are some repetitions when citing papers or equations (e.g. “equation equation [8]“)*
>
> **[Response] Our apologies for the oversight. We've corrected this throughout the manuscript by adjusting our reference format.**
>
> 8. *page 4, The acronym “AD” which probably refers to Automatic Differentiation is not introduced before being used I believe.*
>
> **[Response] You're correct, and we have now introduced the full term where appropriate.**
>
> Thank you again for your insightful feedback. We hope our revisions meet your expectations.
>
> Best regards.

---

> > ### Comment · Reviewer_qiNs · 2023-08-29
> > **Response to response**
> >
> > I thank the authors for taking into account my comments. In particular I appreciate that they included training times and matched the number of training points in their comparisons.
> >
> > My remaining thoughts are:
> > 1. In the example presented in 4.1, the authors claim to have matched the number of training points by doubling it for the single network. Should not this number be multiplied by 5 instead of by 2 since there are 5 networks in the compared GB?
> >
> > 2. While I agree that one cannot make all thinkable comparisons, I also think that Reviewer Naa7 has a good point in stressing the importance of using positional encoding (aka) Fourier features. Table 2, 3, 4 seems to suggest that a significant improvement is acheived when adding the Fourrier network albeit after MLPs. Given also the referees put forward by  Reviewer Naa7, using as a baseline a single network with positional encoding seems to make a lot of sense.

---

> > > ### Author Response · Authors · 2023-08-29
> > > **Response to reviewer**
> > >
> > > Dear Reviewer,
> > >
> > > Thank you for your insightful feedback. Below is a point-by-point response to your queries:
> > >
> > > 1. *In the example presented in Section 4.1, the authors state that the number of training points was doubled for the single network. Shouldn't this number be multiplied by 5 instead of 2, given that five networks are being compared in GB?*
> > >
> > > [Response] I appreciate you bringing this to my attention. Our initial aim in doubling the number of training points was to increase batch size for more accurate gradient descent. We previously mentioned "doubling the batch size" to that end. However, to ensure a more rigorous comparison in terms of the number of training points, we have now increased the training steps to $50,000$ while retaining the original batch size, and updated the result accordingly.
> > >
> > > 2. *While I concur that one cannot perform every conceivable comparison, Reviewer Naa7 makes a valid point regarding the importance of using positional encoding (aka Fourier features). Tables 2, 3, and 4 indicate that considerable improvement is attained when incorporating the Fourier network, particularly after MLPs. Given the concerns raised by Reviewer Naa7, utilizing a single network with positional encoding as a baseline seems quite reasonable.*
> > >
> > > [Response] Reviewer Naa7's key question pertains to whether the improvement in results is SOLELY attributable to the Fourier feature network (“*One could argue that the main advantage of "GB PINNs" is that they contain "Fourier features", which has nothing to do with GB.*”). To address this, our ablation study includes results both with and without Fourier features. Specifically, the penultimate row of Table 2 presents the performance of a standalone Fourier feature network, identical in structure to the one used in the first set of results reported in Section 4.1. This apple-to-apple comparison reveals that using only the Fourier feature network results in an error of $1.27$%, as opposed to an error of $0.43$% when employing GB PINNs that include the same structure of Fourier feature network. To our understanding, this comparison is fair enough and shows the result due to GB PINNs not merely Fourier feature.
> > >
> > > To further address this point, we conducted similar tests for examples presented in Sections 4.2 and 4.3. These new results have been added to Tables 3 and 4 in the ablation study and are highlighted in red. Once again, the standalone Fourier feature networks, identical in structure to those reported in Sections 4.2 and 4.3, demonstrate a significant performance gap compared to GB PINNs (i.e. $15.56$% v.s. $1.03$% and $16.15$% v.s. $3.37$%). In addition, Appendix A.2 has been updated with a paragraph elaborating on these points.
> > >
> > > I hope this adequately addresses your concerns.
> > >
> > > Best regards.

---

> > > > ### Comment · Reviewer_qiNs · 2023-08-30
> > > > **Response**
> > > >
> > > > Thank you very much for your reactivity and the additional experiments.
> > > >
> > > > Concerning the added Fourier networks, they are indeed identical in structure to the one involved in the collection of networks of the corresponding GB PINNs strategy. The experiments demonstrate that conducting a GB procedure can improve significantly the results compared to using the network alone. One last comment, that could provide further insights for readers, if the authors are willing to go as far: following point 1. of my initial review, it would also make sense to use a larger Fourier fetaure network to see whether the gap with the GB method can be filled by using a comparable number of parameters in the single network.
> > > >
> > > > I leave it to the authors to decide if they want to include this comparison.
> > > >
> > > > Best regards.

---

> > > > > ### Author Response · Authors · 2023-08-30
> > > > > **Response to reviewer**
> > > > >
> > > > > Thank you for your comments.
> > > > > I think the Fourier feature is point 2 in your second comment if I understand correctly.
> > > > > In response to your request, we have expanded our analysis to include larger Fourier feature results. These additional findings have been incorporated into Appendix 2, specifically in Table 2-4. We appreciate your guidance on this matter.

---

### Review · Reviewer_Qc7A · 2023-08-15

**Summary Of Contributions:**

This paper proposes to combine gradient boosting (GB) with physics-informed neural networks (PINNs) to overcome some limitations of standard PINNs. This approach is demonstrated on example problems that are known to be difficult for standard PINNs.

**Audience:**

No

**Claims And Evidence:**

No

**Requested Changes:**


There are a number problems the manuscript that I would want to see addressed before publication:
- The text says that at each boosting iteration, the new PINN is trained using gradient descent on the resulting loss function defined in Equation 8. In order to compute this loss, one must compute the gradients through the previous PINNs. However, in Section 3 it is said that "backward gradient propagation is only performed on theta_i", making me question whether I understood the proposed algorithm. Is the algorithm truly performing gradient descent on the PINN loss?
- There were some problems with the logic in the text:
  - In Section 1, three of the four "main contributions" listed are not contributions but rather alleged advantages of the proposed algorithm. Number 2, "increased flexibility for solving intractable problems," is imprecise. Number 3, "Low sensitivity to the choice of neural networks" is not supported by arguments or evidence.
  - In Section 2.2 Equation 6 defines h_m in terms of the residuals, but in the text h_m is defined as the function computed by the neural network. The manuscript should be more clear that the neural network is trained to approximate the residuals, but they are not identical.
  - In Section 3, "In PINNs scenarios, once the training of f_m-1 is complete, we already have a decent predictor, which implies that the relative error between the current predictor and the ground truth is assumed to be small." This is unclear.
  - Just below, "...it is also reasonable to assume that rho_m gradually decreases over m." The learning rate is a hyperparameter, so there is no reason to assume anything about it.
- The manuscript does not provide a good argument for why GB can help solve the limitations of PINNs, other than pointing towards the general success of GB. The manuscript claims that the failures of PINNs are poorly understood (Section 3), but one can construct simple scenarios in which PINNs fail for well-understood reasons (e.g. because they are inefficient at propagating the physics information to regions away from the boundary points), and I see no reason why GB would be helpful in overcoming these challenges.
- In the experiments, it appears that the superior performance is due to ignoring the physics information in favor of fitting the boundary points. Nothing in the description of the experiments indicated that the authors appreciated this distinction.

**Strengths And Weaknesses:**

Strengths:
- Improving PINNs is of interest to the community.

Weaknesses:
- Some key parts of the text are unclear.
- Important details are missing from the experiments.
- While the basic idea of combining two powerful methods (PINNs and GB) is interesting, the paper does not provide any compelling argument for why GB would improve PINNs.
- Some claims are made without supporting evidence.

---

> ### Author Response · Authors · 2023-08-25
> **Response to the reviewer (1/2)**
>
> Dear reviewer,
>
> Thank you for your insightful feedback. We genuinely appreciate the opportunity to enhance our manuscript based on your experienced insights. For clarity, we've marked the changes in the revised manuscript in red. Please note that some of these revisions also respond to comments from other reviewers. Below are our responses to your comments (since the response is limited by 5,000 characters, we did not cite the full comments):
>
> 1. The text says that at each boosting iteration, the new PINN is trained using gradient descent on the resulting loss function defined in Equation 8. In order to compute this loss, one must compute the gradients through the previous PINNs. However, in Section 3 it is said that "backward gradient propagation is only performed on $theta_i$", making me question whether I understood the proposed algorithm. Is the algorithm truly performing gradient descent on the PINN loss?
>
> [Response] We regret any confusion our initial presentation may have caused. In Section 2, the statement "at each boosting iteration, the new PINN is trained using gradient descent on the resulting loss function defined in Equation 8. To compute this loss, gradients through the previous PINNs are required" was made in the context of introducing the GB method for traditional machine learning. However, in Section 3, where we discuss our proposed GB PINNs, we specify that "backward gradient propagation is only performed on $theta_i$."
>
> To provide more clarity: In Section 3.1, we adapted the original GB approach to address certain complexities with gradient computation, as delineated in the initial part of the section. As a solution, we opted for a neural network in place of directly computing the challenging gradient. Consequently, this neural network is then trained via the PINNs process, with $theta_i$ being its specific parameter.
>
> To improve the manuscript's clarity, we've supplemented Section 3.1 with an expanded explanation and emphasized the distinctions between conventional GB and its application within PINNs at the section's conclusion. It's worth noting that this revision is interconnected with the changes highlighted in response 2.2.
>
> 2. There were some problems with the logic in the text:
> 2.1 In Section 1, three of the four "main contributions" listed are not contributions but rather alleged advantages of the proposed algorithm. Number 2, "increased flexibility for solving intractable problems," is imprecise. Number 3, "Low sensitivity to the choice of neural networks" is not supported by arguments or evidence.
>
> [Response] We recognize the inaccuracies in presenting our contributions and have rephrased this section to better summarize our work.
>
> 2.2 In Section 2.2 Equation 6 defines $h_m$ in terms of the residuals, but in the text $h_m$ is defined as the function computed by the neural network. The manuscript should be more clear that the neural network is trained to approximate the residuals, but they are not identical.
>
> [Response] This is absolutely a good point, and we recognize that our initial explanation about this wasn't clear. For $h_m$, we decided to use a neural network rather than directly computing the gradient of the residual with respect to the model, due to the complexity of the computation. As noted in response 1, we have made revisions to Section 3 to highlight this aspect. Furthermore, for better clarity, we have also restructured Section 3.
>
> 2.3 In Section 3, "In PINNs scenarios, once the training of $f_{m-1}$ is complete, we already have a decent predictor, which implies that the relative error between the current predictor and the ground truth is assumed to be small." This is unclear.
>
> [Response] Thank you for highlighting this. What we were trying to convey is that even when PINNs might not produce the best results, they still gain some knowledge during training. We can then employ an additional network to enhance these results. We've addressed this by revising the section discussing the learning rate $\rho_m$. Kindly refer to the updated Section 3.2.
>
> 2.4 Just below, "...it is also reasonable to assume that $\rho_m$ gradually decreases over $m$." The learning rate is a hyperparameter, so there is no reason to assume anything about it.
>
> [Response] You're right. What we meant is that, in most cases, this is what we observe. It's an empirical observation, similar to how we tune hyperparameters. We've made clarifications in Section 3.2 to reflect this.

---

> > ### Author Response · Authors · 2023-08-25
> > **Response to the reviewer (2/2)**
> >
> > (This part proceeds to the previous response)
> >
> > 3. The manuscript does not provide a good argument for why GB can help solve the limitations of PINNs, other than pointing towards the general success of GB. The manuscript claims that the failures of PINNs are poorly understood (Section 3), but one can construct simple scenarios in which PINNs fail for well-understood reasons (e.g. because they are inefficient at propagating the physics information to regions away from the boundary points), and I see no reason why GB would be helpful in overcoming these challenges.
> >
> > [Response] We apologize for not making this clear in our original manuscript. In the revised version, we have added section 3.4 to provide more clarity. First, while GB PINNs differ from traditional GB, they both follow the same principle of using a new model to improve upon the old one. This gives us a reason to believe that GB PINNs might perform better than traditional PINNs. Second, in our discussions in the paper, we explain how GB PINNs tackle issues like invariance of NTK and how they prevent underfitting for both large and small networks. These aspects further argue in favor of GB PINNs.
> >
> > When we say “failures of PINNs are poorly understood”, we are referring to a general lack of understanding. While some specific failure modes have been discussed in works like Wang et al. (2021; 2022) and Krishnapriyan et al. (2021) (cited in the paper), and as you pointed out in your comment, it is not always clear why PINNs sometimes fall short in solving PDEs. We touch upon this in the beginning of section 3 and also in the final paragraph of section 3.4. Our method does not come with strict prerequisites, and our experiments have primarily looked into specific problems like boundary layer issues and a certain nonlinear time evolution problem. We have also noted in our conclusion that our approach is not a solution for conservation law problems.
> >
> > 4. In the experiments, it appears that the superior performance is due to ignoring the physics information in favor of fitting the boundary points. Nothing in the description of the experiments indicated that the authors appreciated this distinction.
> >
> > [Response] Thank you for raising this concern. While we did present results on some boundary layer problems, we have not implemented any specialized handling for the boundary points. The number of boundary points in our method is greater than in traditional PINNs. However, this is due to our multiple training procedures, each having its own set of training points. The size of these sets remains consistent across all GB PINNs network variations. As such, we have not added an additional quantity of boundary points nor have we modified the boundary treatment in our algorithm. Furthermore, we have retained all relevant physics information during the training process. We hope this clarifies the reviewer's query. If there are further questions or any aspect remains unclear, we kindly ask the reviewer to provide more specific details, and we will be eager to address them.
> >
> > Thank you again for your valuable feedback. We hope our revisions meet your expectations.
> >
> > Best regards.

---

> ### Comment · Reviewer_Qc7A · 2023-08-29
>
> Thank you for clarifying some of the details.
>
> If the approach is not suitable for conservation law problems, why bother using PINNs? Perhaps this is a miscommunication, but I think the  only reason use a PINN is to incorporate some physical constraint to help generalize to regions where there is no data. In your experiments, the models appear to be trained on lots of data, with no real need for the physical constraints, so I would expect any high-capacity model to do well.
>
> I suspect that the proposed method would perform poorly on the types of problems that PINNs were proposed for, in which the model needs to use physics to extrapolate far from the boundary points. My reasoning is that each new PINN added to the ensemble will need to try to fix the incorrect gradients of the current ensemble, which could be noisy. I would have liked to see both (1) justification for why this is not a concern, and (2) experiments showing that the proposed method can indeed extrapolate beyond the boundary points.

---

> > ### Author Response · Authors · 2023-08-29
> > **Response to reviewer**
> >
> > Dear Reviewer,
> >
> > Thank you for your comments and questions. Allow me to address your concerns point-by-point.
> >
> > 1. *If the approach is not suitable for conservation law problems, why bother using PINNs?*
> >
> > Your query seems to imply that PINNs are only relevant for conservation law problems, which is not the case. Physics-Informed Neural Networks (PINNs) are versatile and can be applied across a wide range of physical systems, not just those governed by conservation laws.
> >
> > 2. *The only reason to use a PINN is to incorporate some physical constraint to help generalize to regions where there is no data.*
> >
> > In the realm of solving PDEs through machine learning, there are generally two approaches: data-informed and physics-informed. Data-informed methods rely on simulation data, typically derived from other software, to train a neural network. On the other hand, physics-informed approaches, like the one employed in our study, only utilize PDEs and boundary conditions as inputs. These are not treated as 'data' per se, as they don't incur the same computational or acquisition costs as simulation data. During training, we randomly sample points and enforce the satisfaction of the PDEs and boundary conditions, even though no explicit data are provided for these points.
> >
> > 3. *In your experiments, the models appear to be trained on lots of data, with no real need for the physical constraints, so I would expect any high-capacity model to do well.*
> >
> > This assumption is incorrect. Our models are trained exclusively using the PDEs, boundary conditions, and a set of randomly generated points, not on any "lots of data" as you mentioned. Moreover, our experimental results show that high-capacity models do not necessarily perform well under these conditions.
> >
> > 4. *I suspect that the proposed method would perform poorly on the types of problems that PINNs were proposed for, in which the model needs to use physics to extrapolate far from the boundary points. My reasoning is that each new PINN added to the ensemble will need to try to fix the incorrect gradients of the current ensemble, which could be noisy.*
> >
> > It appears that there's a misunderstanding about the methodology we employed. To clarify, our approach also relies solely on PDEs, boundary conditions, and randomly generated points, similar to conventional PINNs. We are not attempting to 'fix incorrect gradients'; rather, our focus is on refining the solution itself. Regarding the concern about 'noise,' it's worth noting that all SGD-class optimizers inherently contain some level of noise, yet they are widely used in various applications without issue.
> >
> > I hope this clarifies any misunderstandings and addresses your concerns adequately.
> >
> > Best regards.

---

### Review · Reviewer_Naa7 · 2023-08-24

**Summary Of Contributions:**

This paper proposes a method to improve the accuracy of Physics Informed Neural Networks (PINNs). To summarize, this method consists in augmenting sequentially the initial neural network (NN), following the idea behind the "Gradient Boosting" (GB) approach. In practice, at each training step, a new NN is added to the model, in order to compensate the error made by this model. So, progressively, the model size increases, while providing more accurate results.

**Audience:**

Yes

**Claims And Evidence:**

No

**Requested Changes:**

Above all, I would like clarification about: how the sequence of NNs used in PINNs is chosen and the value of $m$ in each case.

**Strengths And Weaknesses:**

# Strengths

Apparently, the authors have managed to increase greatly the performance of PINNs by using GB, especially where the solution of the PDE has high frequency components.

However, the experimental setup is far from being clear for me. So I would like some explanations about it (see below).

# Weaknesses

## Formatting

The citations are badly handled (many citations contain the name of the author twice).

## Notation

Variables $m$, $i$, $k$ are used inconsistently. In Section 2.2, the training step index of GB is denoted by $m$. In Algorithm 1, $m$ becomes $i$, and $m$ is now used as the number of training steps. Finally, in Section 4.1, we read "As we advance to training the $k$-th network [...]", which is not consistent with the preceding notation.

## Clarity: sequence of NNs used

In 4.1, the authors write that:
> We used a series of fully connected network structures sequentially (1, 50, 1), (1, 100, 1), (1, 100, 100, 1), (1, 100, 100, 100, 1) for the baseline and update neural networks, followed by a Fourier feature neural network (1, 50, 50, 1) [...]

I understand the notation (1, 100, ..., 1) to describe the architecture of each NN, but I do not understand how they are used in GB PINNs: A) do we have $m = 5$ GB steps with one specific architecture per step, i.e., sequentially (1, 50, 1), etc.? B) did the authors test each one of the 4 architectures (followed by a Fourier feature NN), with various $m$?

In case B, the value of $m$ should be reported somewhere, for each one of the architectures.

In case A, I do not understand why the "model index" $i$ can take $6$ values, i.e., $0, 1, \cdots, 5$, while only $5$ NNs are presented. Besides, I do not understand the following statement:
> The peak memory requisition touched 0.28GB. In a scenario where only the largest configuration within the GB PINNs’ network spectrum, specifically (1, 100, 100, 100, 1), is utilized, the memory footprint scales down to 0.16GB.

If we use 5 times the NN of architecture (1, 100, 100, 100, 1) instead of the series presented above, more memory should be used, right?

Overall, I did not understand the experimental setup.

# Questions

## "The neural network architectures employed include (2, 200, 200, 200, 1), (2, 100, 100, 100, 1), (2, 100, 100, 1), and a Fourier feature network (2, 50, 50, 1)"

The word "include" is ambiguous.

What does it mean? Are these architectures randomly selected at each GB step $m$?

## Comparison with a simple ensemble method

The authors use a kind of ensemble method by using GB. I would like to know the performance of an ensemble of NNs, with the same architecture as in GB PINNs, and the same number $m$ of NNs, when trained by standard SGD. I assume that it would lead to slightly better results, but will be longer to train (since the NNs of the ensemble would be trained simultaneously).

Such a comparison would provide a computational argument in favor of GB PINNs.

## Ablation study: larger ensembles do not lead to better performance

In tables 4 and 5, it is striking that the richest ensembles do not lead to the best performance. Is there an explanation to that?

---

> ### Author Response · Authors · 2023-08-27
> **Response to reviewer**
>
> Dear Reviewer,
>
> Thank you for your insightful comments. We appreciate the time and effort you took to review our manuscript. To address your points, we have made revisions and marked these changes in red for clarity. Please note that some of these revisions also respond to comments from other reviewers. Below are our detailed responses to your specific remarks:
>
> 1. *Formatting & Notation*
>
> [Response] We appreciate your attention to detail. All citations and notations have been revised in accordance with your recommendations.
>
> 2. *Clarity: sequence of NNs used*
>
> [Response] We apologize for the confusion surrounding choice (A) as described in the paper. As specified in Algorithm 1, we preselected $f_0$ and $h_m,$ and set $f_m=f_{m-1}+\rho_m h_m$, where m is the step index of GB. We are grateful for your observation about inconsistent notations. In the output part of the algorithm, GB steps from 0 to M are clearly delineated.
>
> *In case A, I do not understand why the "model index" I can take 6 values, i.e., 0,1,…5, while only 5 NNs are presented.*
>
> [Response] The model index range from 0 to 5 was a typographical error; the correct range should be 0 to 4. This has been corrected in the revised manuscript.
>
> *If we use $5$ times the NN of architecture $(1, 100, 100, 100, 1)$ instead of the series presented above, more memory should be used, right?*
>
> [Response] In reality, a $(1, 100, 100, 100, 1)$ network comprises $20,501$ variables. Given that these variables are float32, the memory requirement would be 82 KB, well within our hardware capabilities. The stated 0.16 GB includes not just the model but also the training points and gradient data stored on the GPU.
>
> Additionally, to enhance clarity, we have rephrased the text in section 4.1 to ensure that the notation aligns consistently with that in Algorithm 1. We have also added a note to indicate that simplified terminology will be employed in subsequent sections to describe the network architectures.
>
> 3. *The neural network architectures employed include…*
>
> [Response] Thank you for highlighting this ambiguity. The word "include" could imply that the order of the networks does not matter, which is not the case. We have rephrased this section in all four numerical experiments to emphasize the sequence in which GB trains the networks.
>
> 4. *Comparison with a simple ensemble method*
>
> [Response] Thank you for your insightful suggestion. To respond to your query, we have incorporated an additional experiment at the end of Section 4.1. In this experiment, we utilize the same neural network architectures as specified at the beginning of Section 4.1. The networks are trained for a sufficient number of steps to ensure adequate exposure to the training data.
>
> While the results indicate a reasonably low relative l2 error, a detailed inspection reveals notable anomalies at the boundary layer of the solution. Specifically, pronounced jumps occur at these boundaries, which compromises the integrity of the overall solution. Thus, while the ensemble approach using standard SGD might yield marginally better results in terms of l2 error, it also introduces distortions that are absent in our GB PINNs framework. This lends computational credence to the efficacy of GB PINNs as a more robust approach.
>
> 5. *Ablation study: larger ensembles do not lead to better performance*
>
> [Response] We appreciate your keen observation. To elucidate this phenomenon, we have added a paragraph at the end of Section 4.1. Essentially, the error distribution becomes less smooth after several GB steps, making it a challenging task for additional neural networks to improve performance.
>
> 6. *Above all, I would like clarification about: how the sequence of NNs used in PINNs is chosen and the value of in each case.*
>
> [Response] Thank you for posing this thought-provoking question. Our choice of neural network architectures for the numerical experiments is informed by insights from the neural tangent kernel. We generally begin with smaller networks and then incrementally introduce larger ones. While this approach is not universally optimal, no definitive strategy exists for identifying the best network combination in GB PINNs. We discuss this limitation in the concluding section, highlighting it as an avenue for future research.
>
> Thank you once again for your valuable feedback. We hope that our revisions meet your expectations and look forward to your further comments.
>
> Best regards.

---

> > ### Comment · Reviewer_Naa7 · 2023-08-28
> > **Usage of Fourier Features method**
> >
> > Thank you for your answer. Now, I think I understand the experimental setup. So, I have more questions about it.
> >
> > ## Testing GP PINNs against simple PINNs *with Fourier features*
> >
> > A major issue in the experimental evaluation of GP PINNs seems to be the absence of Fourier features in the standard PINNs against which the authors test their method. One could argue that the main advantage of "GB PINNs" is that they contain "Fourier features", which has nothing to do with GB. So, GB PINNs have to be tested against PINNs with Fourier features.
> >
> > ## How to use Fourier features?
> >
> > According to [1] and [2], Fourier features is mainly an augmentation of the input data points: instead of sending a input point $\mathbf{x}$ to a network, one sends $(\cos(\mathbf{a}_i \cdot \mathbf{x}), \sin(\mathbf{a}_i \cdot \mathbf{x}))_i$ for a given number of "frequencies" $(\mathbf{a}_i)_i$. The initial input $\mathbf{x}$ could even be added to this large vector of $\cos$ and $\sin$. The resulting input is believed to be more manageable to perform 2D reconstruction or solve PDEs, and this is supported by series of experiments.
> >
> > However, in the present setup of the paper, Fourier features are used in a different way, that looks very strange to me. In GB PINNs, Fourier features are used to match the residuals *only at the final GB step*. Why?
> >
> > ## Proposition of an experimental setup
> >
> > If we take seriously the setup of [1] and [2], we should forget completely the idea of sending raw inputs $\mathbf{x}$ to any MLP. One should pre-process $\mathbf{x}$ first to extract its Fourier features. Possibly, the vector of Fourier features could be augmented with raw $\mathbf{x}$ itself. The resulting vector, that we denote by $\tilde{\mathbf{x}}$, could then be used (1) in the GB PINNs setup, (2) in the standard PINN setup, and (3) in the "ensemble of PINNs" setup. That way, the authors could test their method (1) against a state-of-the-art method (2) and a method simpler than (1), but very close to it.
> >
> > In practice, it would mean that the authors replace $\mathbf{x}$ by $\tilde{\mathbf{x}}$ everywhere, and that they remove the "Fourier features NN" part from their setup.
> >
> > ## References
> >
> > [1] *Fourier Features Let Networks Learn High Frequency Functions in Low Dimensional Domains*, Tancik *et al.*, 2020.
> >
> > [2] *Physics-Informed Neural Network with Fourier Features for Radiation Transport in Heterogeneous Media*, Huhn *et al.*, 2023.

---

> > > ### Author Response · Authors · 2023-08-28
> > > **Response to reviewer**
> > >
> > > Thank you for your constructive comments. Below is a point-by-point response to your queries.
> > >
> > > 1. *Testing GP PINNs against Simple PINNs with Fourier Features*
> > >
> > > [Response] Indeed, in Sections 4.3 and 4.4 of our experiments, Fourier features were not employed. Our ablation study comprehensively examines different network architectures, including those both with and without Fourier features. We believe this provides a robust evaluation. Fourier features may enhance results by smoothing the neural tangent kernel, but it would be inaccurate to assert that the primary advantage of GP PINNs lies solely in their use of Fourier features. As you pointed out, Fourier features are used at the final step thus weighted least in our architecture, and they primarily serve to address high-frequency errors. Tables 2 through 5 further demonstrate that while Fourier features can notably enhance results in some instances, they are not the sole contributors to the observed performance improvements. In Table 2, a Fourier-features-only model using the same architecture as in Section 4.1 actually yielded inferior results.
> > >
> > > 2. *How to Use Fourier Features?*
> > >
> > > [Response] Our approach to employing Fourier features is clearly outlined at the outset of Section 3 (fifth line of the first paragraph). We base the use of Fourier features on empirical observations and existing theoretical frameworks. As noted in the paper, there is currently no universally agreed upon methodology for optimizing network structures. In such a scenario, we rely on informed observations and, at times, educated guesses, as discussed at the beginning of Section 3.
> > >
> > > 3. *Proposition of an Experimental Setup*
> > >
> > > [Response] To reiterate, we do not ubiquitously deploy Fourier features in our model. Our implementation of Fourier features involves positional encoding followed by a Multi-Layer Perceptron (MLP), a construct supported by extensive literature. We employ this Fourier feature network exclusively to address high-frequency errors. Regarding your suggestions for additional experimental setups, it appears you are advocating a fusion of positional encoding and GP PINNs. While intriguing, we are committed to maintaining a conservative scope for our research. Our primary focus is on comparing our method against well-established methodologies rather than inventing a multitude of new frameworks for comparative analysis. Firstly, these newly created methods have not been independently verified, casting doubt on their suitability for a fair comparison. Secondly, pursuing such an approach would make the comparison process interminable, given that existing structures could be endlessly modified or combined to create new configurations.
> > >
> > > Therefore, we opt to maintain a conservative focus for our study, strictly comparing our method with existing, recognized models.
> > >
> > > We hope that our revisions meet your expectations and look forward to your further comments.
> > >
> > > Best regards.

---

> > > > ### Comment · Reviewer_Naa7 · 2023-09-04
> > > >
> > > > I appreciate greatly that the authors have included experiments with simple PINNs with Fourier features in their experiments, despite the results are still in Appendix.
> > > >
> > > > Overall, I still have serious doubts about the actual contributions of the paper and its scope:
> > > >  1. the paper intends to prove the usefulness of GB PINNs.
> > > >  2. I think that the actual contribution is: in order to have PINNs with good performance, one has to create an ensemble made of standard MLPs and MLPs with Fourier features. One may train this ensemble with a GB method to increase the training speed (still to demonstrate).
> > > >
> > > > I consider that the main claim 1 is not strongly proven:
> > > >  * according to Tables 2, 3, some large GB PINNs without Fourier features are outperformed by a single MLP with Fourier features, showing that Fourier features is (at least) as important as GB PINNs themselves. Thus, focusing the paper on GB PINNs does not represent faithfully the actual experimental results;
> > > >  * despite the importance of Fourier features, these is no reference to it in the list of contributions;
> > > >  * the theoretical support for using GB in this situation (presented in Section 3.4) is very weak:
> > > >     * the argument "GB, a recognized technique in traditional machine learning tasks like tree models" is not relevant in the current situation. We do not deal with trees, and using GB when training NNs is not "a recognized technique". Actually, I believe this is rarely used in NNs. I could accept this argument only the authors provide a reference to a paper where GB is successfully applied to ensembles of MLPs,
> > > >     * the intuition behind GB is interesting and should be kept, but it is only an intuition, and not a fact or an argument in favor of GB PINNs,
> > > >     * "Secondly, as emphasized [...]": this argument advocates for ensembles (and nor GB specifically). Also, the argument is based on an analysis made is the NTK/lazy-training regime, which is a bit extreme (layers with an infinite number of neurons). If this argument holds, then nobody would use large NNs, which is not the case. If the authors think that there is a "lazy training" problem in their experiments with simple MLPs, they should provide an experimental evidence of it.
> > > >
> > > > However, the actual experiments provide some evidence that, for physical problems:
> > > >  * ensembles of small NNs work better than large NNs;
> > > >  * adding at least one MLP with Fourier features increases greatly and consistently the performance;
> > > >  * one MLP with Fourier features, even a large one, may fail in some circumstances (Table 4).
> > > >
> > > > These empirical results are interesting in themselves, and could be put in the main text. But one has to find more arguments in favor of GB, compared to a simple ensemble training.
> > > >
> > > > To summarize, I think that the authors focused too much on GB, provided the experimental results and intuitions given in Section 3.4. With a slight change of focus, some empirical results, convincing, useful for the practitioner and based on experimental evidence, could be highlighted. Notably, the "ablation study" is of great interest, and the importance of Fourier features should be highlighted (along with its limitation, see Table 4).

---

> > > > > ### Author Response · Authors · 2023-09-05
> > > > > **Response to reviewer**
> > > > >
> > > > > Thank you for your insightful comments and suggestions. In consultation with the action editor, we have organized our responses as follows:
> > > > >
> > > > > 1. **Fourier Features**
> > > > >
> > > > > We appreciate your keen interest in Fourier features. While it's true that these features significantly enhance the performance of PINNs, it's important to note that this is not a novel contribution of our work. Fourier features have been rigorously studied in prior literature, including publications by Sifan Wang (2021, 2022) and Matthew Tancik (2020), among others. Our work leverages Fourier features to improve the efficacy of our proposed GB PINNs model, but we do not consider them to be on the same tier of contribution. To offer an analogy, in transformer architectures, dropout and residual connections, although crucial, are not typically listed alongside attention mechanisms as primary contributions. Nonetheless, to address your concerns, we have included Fourier features in the "Contributions" section, as indicated in red in the revised manuscript.
> > > > > If the reviewer is interested in Fourier features, we refer them to our cited references as well as the references within those works.
> > > > >
> > > > > 2. **GB and Ensemble Approaches**
> > > > >
> > > > > Your comment about an excessive focus on GB is noted. However, GB is the central element of this paper, as also indicated in the title. The purpose of GB in this context is not to accelerate training but to enhance model performance. Furthermore, in our work, GB and ensemble methods are intrinsically linked, paralleling traditional GB approaches.
> > > > >
> > > > > Regarding your observation that *"GB, a recognized technique in traditional machine learning tasks like tree models, is not relevant in the current situation,"* we believe there may be some misunderstanding. The phrase "recognized technique" serves as a descriptor for "traditional machine learning tasks," illustrated by the example of "tree models." We did not imply that using GB in neural networks is universally recognized. The mention of tree models was merely to trace the origins of GB methods and was not intended to be directly relevant to this paper.
> > > > >
> > > > > 3. **Lazy Training**
> > > > >
> > > > > We agree that the point on lazy training is not extreme. Lazy training and neural tangent kernels are established theories described in existing literature, which we have cited to provide foundational understanding. Our work references these theories to elucidate our conceptual framework. As for the question of infinite depth and width in neural networks and the subsequent assertion that "then nobody would use large NNs," we defer to the seminal papers that introduced these concepts.
> > > > >
> > > > > In summary, we have thoughtfully incorporated your comments about Fourier features into the revised manuscript. We look forward to receiving your further feedback.
> > > > >
> > > > > Best regards.

---

### Decision · Action_Editors · 2023-09-24

**Recommendation:** Reject

**Comment:**

After careful consideration and discussion, the majority of reviewers agreed to reject this paper. While the paper contains many interesting insights regarding training PINNs, many of them are hidden in the ablation studies in the appendix. Overall, the paper shows that both using Fourier features (as done in previous work) as well as ensembling can improve performance in PINNs. It also shows that gradient boosting can have some benefits over standard ensembles, for instance at boundary points, but it is not clear that it is generally better. The reviewers and AE would recommend toning down the claims about GB and turning the paper into a more honest discussion and evaluation of different PINN techniques in different settings (by moving the results from the appendix into the main text). The authors should show on different physical problems how single vs. ensemble vs. GB PINNs perform, both with and without Fourier features. It may turn out that GB PINNs will indeed perform the best on some problems, but possibly not in terms of all metrics. Also, the theoretical claims about lazy training should be empirically tested.
If such a major revision were made, we would welcome a resubmission of the paper.

**Audience:**

The problem of improving PINNs is very important and of interest to at least some of TMLR's audience.

**Claims And Evidence:**

The current claims made in the paper, namely that gradient boosting in PINNs leads to a wide range of benefits, are not entirely backed by evidence. Namely, some benefits, such as mean accuracy, can also be achieved by standard deep ensembles, while others, such as the robustness to "lazy training" problems, are never empirically tested.

**Resubmission Of Major Revision:**

The authors may consider submitting a major revision at a later time.